# Attempted Control of Paratuberculosis in Dairy Calves by Only Changing the Quality of Milk Fed to Calves

**DOI:** 10.3390/ani11092569

**Published:** 2021-08-31

**Authors:** Pamela Steuer, Carlos Tejeda, Manuel Moroni, Cristobal Verdugo, Michael Thomas Collins, Miguel Salgado

**Affiliations:** 1Laboratorio de Enfermedades Infecciosas, Instituto de Medicina Preventiva Veterinaria, Facultad de Ciencias Veterinarias, Universidad Austral de Chile, Casilla P.O. Box 567, Valdivia, Chile; pamelasteuer@gmail.com (P.S.); carlostb81@gmail.com (C.T.); cristobal.verdugo@uach.cl (C.V.); 2Escuela de Graduados, Facultad de Ciencias Veterinarias, Universidad Austral de Chile, Casilla P.O. Box 567, Valdivia, Chile; 3Instituto de Patología Animal, Facultad de Ciencias Veterinarias, Universidad Austral de Chile, Casilla P.O. Box 567, Valdivia, Chile; manuelmoroni@uach.cl; 4Department of Pathobiological Sciences, School of Veterinary Medicine, University of Wisconsin, Madison, WI 53713, USA; michael.t.collins@wisc.edu

**Keywords:** MAP, copper, milk, calves, infection transmission

## Abstract

**Simple Summary:**

Milk is an important transmission route of *Mycobacterium avium* subsp. *paratuberculosis* (MAP) for dairy calves. The aim of the present study was to assess the efficacy of milk treatments to prevent MAP infection transmission to calves. A one-year field longitudinal study was carried out. Newborn calves were assigned to one of four experimental groups. MAP numbers were estimated for treated and untreated milk. Infection progression was monitored monthly. After one year, calves were euthanized, and tissue samples examined grossly and by histopathology. Milk treatments were highly effective though MAP shedding was also observed in all groups. It was not enough to try to block MAP infection transmission controlling only the quality of milk consumed by calves.

**Abstract:**

One of the important routes of *Mycobacterium avium* subsp. *paratuberculosis* (MAP) transmission in dairy calves is milk. The aim of the present study was to assess the efficacy of milk treatments to prevent MAP infection transmission to calves. A one-year longitudinal study was carried out. Newborn calves were assigned to one of four experimental groups: 5 calves received naturally MAP-contaminated milk, 5 calves received copper treated milk, 4 calves were fed calf milk replacer, and 3 were fed UHT pasteurized milk. MAP load in milk was estimated. Infection progression was monitored monthly. After one year, calves were euthanized, and tissue samples were cultured and visually examined. MAP was undetectable in milk replacer and UHT milk. Copper ion treatment significantly reduced the number of viable MAP in naturally contaminated milk. Fecal shedding of MAP was observed in all study groups but began earlier in calves fed naturally contaminated milk. Paratuberculosis control programs must place multiple hurdles between the infection source, MAP-infected adult cows, and the most susceptible animals on the farm, young calves. As our study shows, strict dependence on a single intervention to block infection transmission, no matter how important, fails to control this insidious infection on dairy farms.

## 1. Introduction

The pathobiology of *Mycobacterium avium* subsp. *paratuberculosis* (MAP) infections is complex. Animals, mainly ruminants, become infected at an early age, neonates being the most susceptible [1,2]. This obligate intracellular pathogen invades and replicates within macrophages [3]. MAP evades the host immune system leading to a chronic progressive infection with a long incubation period [1,3].

The infection remains subclinical for 2 to 7 years or longer in cattle [2,3,4]. Clinical signs in dairy cattle include weight loss, intermittent diarrhea, cachexia, and decreased milk production. Prior to onset of clinical signs, MAP-infected cattle shed MAP bacteria in their feces and milk, acting as sources of infection for susceptible individuals through contamination of colostrum and milk fed to calves as well as the farm environment, thereby perpetuating the infection cycle [5,6]. During the incubation period, the available diagnostic tests have high rates of false-negative results (low diagnostic sensitivity), making identification of MAP-infected adult cattle, and thus infection control, on farms challenging and hampering national control programs [7].

Explanations for the continued spread of paratuberculosis globally in dairy cattle are complex but ultimately are related to the difficulty of interrupting the MAP transmission cycle due to three main factors: the pathobiology of the infection, the management of newborn calves in dairy herds, and characteristics of MAP, notably its resistance to adverse environmental conditions. Consolidation of the dairy industry leads to fewer and larger herds. There is a significant association between herd size and herd-level prevalence of paratuberculosis in dairy herds [8].

Several routes of infection have been proposed for MAP transmission: fecal–oral, through mammary secretions, in utero, and via semen. However, the most important route of MAP transmission in all species is thought to be fecal–oral, i.e., by ingestion of fecal-contaminated solid feed or water or through nursing from an infected dam via fecal-contaminated teats, or by direct shedding of MAP into colostrum or milk [9,10,11].

MAP has been isolated from milk and colostrum of cows during subclinical and clinical stages of infection [12,13] and if teats are not properly cleaned and disinfected prior to harvest, colostrum and milk may become MAP-contaminated during milking [7]. Therefore, it is important to eliminate MAP in milk fed to calves to improve the efficacy of paratuberculosis control in dairy herds.

The use of thermic process such as pasteurization have been considered effective in controlling viable MAP in raw milk. However, recent evidence has raised concern regarding the safety of calf milk replacer [14] and the efficacy of pasteurization [15,16]. For this reason, our group has explored an alternative to heat treatments, based on copper ions [17,18], as a means of decreasing microbial loads in raw milk. Therefore, the aim of the present study was to assess the efficacy of milk treatments to prevent MAP infection transmission to calves.

## 2. Materials and Methods

### 2.1. Study Design

A year-long pilot longitudinal field study was carried out to assess the efficacy of milk treatments to prevent MAP infection transmission to dairy calves through naturally contaminated milk. The study was carried out on an experimental animal research facility located in the Los Ríos region, Southern Chile, between November 2017 and October 2018. Seventeen newborn male calves were randomly assigned upon arrival to one of four experimental groups and each animal received a dairy diet of 4 L milk/day for 12 weeks. Group A calves (*n* = 5) received milk naturally contaminated with MAP (collected from 5 infectious nurse cows), and group B calves (*n* = 5) were fed the same MAP-infected milk after first being treated with a novel decontamination tool based on copper ions. As controls for thermic-treated milk, group C calves (*n* = 4) received commercial milk replacer (Sprayfo™ Trouw Nutrition, Boxmeer, The Netherlands), and group D calves (*n* = 3) were fed UHT milk marketed for human consumption (COLUN).

We selected a small sample size considering mainly animal welfare 3Rs, reducing the number of animals used to a minimum.

Only the two main investigators (PS and MS) were aware of the group allocations, but the samples were collected, processed, and analyzed in a blind manner, since the technical staff and the rest of the co-authors did not know the individual identification of each calf, and to which group each one belonged until the data were analyzed.

### 2.2. Study Population

#### 2.2.1. Calves

The seventeen newborn calves were acquired within a month from 3 small dairy herds (herd size between 50 to 80 lactating cows), all located in different districts within the Los Ríos region, Southern Chile, with history of negative paratuberculosis diagnostic results for at least 5 years based on culture and ELISA, and absence of clinical cases. The only exclusion criterion was if the calf had nursed. Calves were immediately separated from their mothers (before they were able to nurse on their own), fed colostrum from their own dam (first day of life) in a hygienic environment, kept in individual pens, and then transferred to the experimental farm (a completely separate farm with monitored hygiene) once calves had a dry hair coat and navel cord, according to international animal regulations regarding animal welfare [19]. Afterwards, as the animals arrived the experimental farm, they were randomly separated into one of four experimental groups as described above, ensuring as a first priority that there were five animals for groups A and B. During the first 12 weeks, the calves were also offered hay, concentrate, and water ad libitum. Finally, from 12 weeks onwards, the animals also received grass silage. The origin of the silage and hay was in the surroundings where the experimental study was carried out. This material is representative of the area, in terms of nutritional quality. The weight of the calves was monitored monthly. Each group of calves was kept in separated pens (without direct contact between groups) until the end of the one-year study period. From the fourth month to the end of the study period, the calves had access to an open yard, while maintaining group separation, which guaranteed the minimum area for each calf for animal welfare, i.e., at least 2.6 m^2^ per animal [20]. Technical staff entered the pens wearing clean coveralls, boots, and gloves used exclusively for each pen. Likewise, each pen had dedicated cleaning equipment to eliminate the risk of cross contamination.

#### 2.2.2. Nurse Cows

Five MAP-infected and infectious lactating cows (confirmed by serum ELISA and fecal culture) were acquired from a herd with a history of high within-herd paratuberculosis prevalence and several clinical cases per year. These cows were selected as nurse cows, transferred to the experimental farm, and the milk from them was pooled fresh and used to feed experimental calf groups A and B. The nursing cows were brought into the experimental farm and were kept only on pastures and milked in a distant milking parlor where the calves were kept at least a distance of 100 m, in order only to allow MAP transmission through milk handling and not to physical proximity.

### 2.3. Milk Treatments

The naturally MAP-contaminated milk was decontaminated using a copper treatment adapted from Steuer et al. [17], modified for field conditions. Briefly, the treatment device consisted of an aluminum receptacle containing 50 L of milk naturally contaminated with MAP in which two high purity copper cylinders were immersed. The copper cylinders were stimulated with a low voltage (24 V) electric current (3 A) to quickly release copper ions for 8 min. The receptacle was carefully shaken during treatment to allow constant mixing.

The milk replacer (Sprayfo™) used for feeding calves from group C was pasteurized and then high-pressure homogenized during its industrial processing, in accordance with the manufacturer’s instructions (Trouw Nutrition, Boxmeer, The Netherlands).

The milk for human consumption (COLUN) used for feeding group D calves, as part of its manufacturing process, was subjected to cooling, pasteurization, and standardization before the UHT (ultra-high temperature) treatment, at 138 °C for 4 s.

#### 2.3.1. Bacteriological Analyses of Milk and Environmental Samples

All types of milk and milk replacer were sampled once a week for a total of 12 weeks (duration of milk feeding), in order to detect MAP and estimate the viable numbers in milk. To achieve this, milk samples were cultured in the BACTEC-MGIT 960 liquid culture system (Becton Dickinson, Sparks, MD, USA). For all milk samples except those treated with copper ions, a decontamination step before the inoculation of liquid culture media was carried out, in accordance with the work of Dundee et al. [21]. Briefly, 50-mL milk samples were centrifuged at 2500× *g* for 15 min and the pellets were resuspended in 10 mL of 0.75% (*w*/*v*) hexadecylpyridinium chloride (HPC, Sigma) and incubated for 5 h at room temperature (20–21 °C). Then, the samples were centrifuged again at 2500× *g* for 15 min and the pellets resuspended in 1 mL of phosphate buffered saline (PBS). Finally, 100 µL of the resuspended pellets were inoculated into MGIT tubes (Becton Dickinson, Sparks, MD, USA) supplemented with 0.8 mL of ParaTB supplement (Becton Dickinson, Sparks, MD, USA), 0.5 mL of egg yolk (Becton Dickinson, Sparks, MD, USA), and 0.1 mL of polymyxin B, amphotericin B, nalidixic acid, trimethoprim, and azlocillin (PANTA) antibiotic mixture (Becton Dickinson, Sparks, MD, USA). Copper-treated milk samples were processed similarly but without the HPC decontamination step and without the addition of PANTA in the culture medium to avoid antibacterial effects other than that caused by copper ions.

MGIT tubes flagged as positive (indicating a change in oxygen concentration in the MGIT tube) were subjected to DNA extraction and purification according to a published protocol [22], and then tested by IS900-qPCR to confirm MAP presence [22]. In addition, direct MAP detection in milk was performed by qPCR and the total MAP load in milk (live and dead cells) was estimated, as explained below.

Along with the bacteriological analyses of milk, hay, concentrate, silage, and water were also cultured in the BACTEC-MGIT 960 system (Becton Dickinson, Sparks, MD, USA) and confirmed by IS900 qPCR. Milk was collected prior to feeding in 50-mL sterile Falcon tubes and cultured once a week during the first 12 weeks. Environmental samples (hay, water, concentrate, and silage) were cultured monthly until the end of the study period. Domestic tap quality water was offered to the animals in hygienic drinking containers. Two to three grams of food and water samples were weighed, suspended in sterile distilled water, vortexed, and then allowed to stand for 30 min. After this time, the supernatant was collected and processed as a sample in the above-mentioned culture system protocol.

#### 2.3.2. Estimation of MAP Load in Milk Samples

Bacterial load (total live and dead cells) from naturally contaminated milk samples before and after copper ion treatment, and from milk replacer and UHT milk were estimated by qPCR (Roche 2.0 real-time PCR) using a standard curve as previously described [23]. Briefly, this standard curve was based on the concentration of MAP DNA measured in a NanoQuant spectrophotometer (TECAN group, Männedorf, Switzerland), adjusted to a 10^8^ dilution, the number of copies of the IS900 target gene, and the reference of the molecular weight of the genome of MAP ATCC strain 19698. The copy numbers of the target region were expressed as MAP-specific bacterial cell equivalents (bce), according to Dzieciol et al. [24].

#### 2.3.3. Evaluation of the Infection Progression in the Study Animals

##### Infectious Status Determination

MAP infection in calves were evaluated before the first milk intake and then monthly both by fecal cultures, using the BACTEC-MGIT system, and serology using an ELISA kit (IDEXX, Westbrook Maine, ME, USA). For culture, 5 to 10 g of fecal material was obtained directly from the rectum using individual palpation sleeves. Blood samples (5 to 10 mL) were obtained by jugular venipuncture up to the first month, and from then on from the coccygeal vein, using Vacutainer™ tubes (BD Diagnostics, Franklin, NJ, USA). All samples were transported to the Laboratorio de Enfermedades Infecciosas, Instituto de Medicina Preventiva Veterinaria, Facultad de Ciencias Veterinarias, Universidad Austral de Chile. Fecal and blood samples were kept at room temperature until processing the following day. The serum was stored at −20 °C.

##### Post-Mortem Analyses

At the end of the study period (12 months), a necropsy was performed on all calves. The pathologist was blind as to treatment group for all animals. Calves were euthanized by a veterinarian pathologist using a retained projectile pistol, in accordance with the criteria of the Bioethics Committee of the Research and Development Department (DID) of the Universidad Austral de Chile (Validation report N° 263–2016). Ileum and lymph nodes samples were cultured in the BACTEC-MGIT 960 system for MAP detection, as previously described.

#### 2.3.4. Ancillary Data: Monitoring Potential Copper Toxicosis in Calves

Copper concentrations in both untreated and copper-treated milk samples were measured by atomic absorption spectrophotometry at each milk sampling during the 12 weeks of feeding. Once the milk diet was completed at 12 weeks of age, and with the aim to confirm or rule out whether a copper accumulation or intoxication effect was produced by the copper treatment of milk, monthly blood samples were used to evaluate copper plasma concentration and the plasma activity of liver enzymes gamma-glutamyl transferase (GGT), glutamate dehydrogenase (GD), and aspartate aminotransferase (AST). In addition, liver samples were collected at necropsy to determine hepatic copper concentrations.

#### 2.3.5. Statistical Analysis

Normality of the data was checked using the Shapiro–Wilk normality test. A comparison of the MAP load (using the log_10_ of the bce estimates) between treated and untreated milk was conducted using the Wilcoxon–Mann–Whitney test. Survival analysis using the Kaplan–Meier estimator [25] was used to assess differences in the time of the first fecal culture positive between groups. Finally, a Bayesian fitted zero-inflated Poisson mixed model (ZIPMM) was fitted to compare the MAP fecal load between groups, where the log_10_ of the MAP bce estimates was rounded to the closest integer (*Y_ijk_*), representing the MAP load (response variable) for the *i*th calf in, in the *j*th group (*X_j_*), at the *k*th sampling time. Control groups (C and D) were defined as the reference for comparison to the groups receiving MAP-contaminated milk (A and B). In general, ZIPMM is used when data are over-dispersed due to an excess of zero counts in the response variable [26]. All statistical analyses were run using the statistical software R version 3.6.3 (R core team 2020). In the particular cases of the survival analysis and ZIPMM, those analyses were run using the packages “survival” [27] and “GLMMadaptive” [28].

## 3. Results

### 3.1. Descriptive

All seventeen calves survived to the end of the study and no differences were seen among study groups in monthly weight increase (data not shown).

### 3.2. Detection of MAP in Milk and Environmental Samples

Only one milk sample, from the untreated naturally MAP-contaminated milk group, showed positive culture results (data not shown). However, by qPCR the average concentration of 3.85 × 10^4^ MAP bce/mL in untreated MAP-contaminated milk was higher than that of the copper-treated milk, in which MAP DNA was detected only on two occasions (Table 1). No MAP was detected in calf milk replacer or UHT milk by culture or PCR. Among environmental samples, only silage and hay showed positive culture results on MGIT media, although this occurred at a low frequency (hay: 1 positive out of 12 samples; silage: 2 positives out of 9 samples).

### 3.3. Detection of MAP Infection in Calves

Culture-positive fecal samples were found at least once in all experimental groups during the study period. However, only in calves fed untreated MAP-contaminated milk (group A) was fecal shedding observed during the first 12 weeks of the study (Table 2). In addition, only among this group of calves were anti-MAP antibodies detected. None of the calves in any study group had positive cultures for MAP from fresh tissues collected at necropsy. However, calves from group A (untreated milk) and group B (copper-treated milk) had gross pathology consistent with a MAP infection, e.g., enlarged lymph nodes adjacent to the last third of the jejunum and ileum (Figure 1A) with reduced medullary area (Figure 1B), lymphangitis in the ileum’s serosa (Figure 1C), and thickening and folding of the ileal mucosa (Figure 1D).

### 3.4. Analytical

A significantly (*p* ≤ 0.01) higher MAP concentration was found in the untreated milk compared to the copper-treated milk that was fed to the calves in groups A and B, respectively. Survival analysis did not show any significant differences (*p* = 0.34) in the time to the first culture positive fecal sample between groups. Similarly, no significant differences in MAP fecal shedding were found between groups by ZIPMM analysis.

### 3.5. Ancillary Data: Monitoring Copper Toxicosis in Calves Fed Copper-Treated Milk

Copper-treated milk showed, on average, a copper concentration 12 times higher than that of untreated milk (1.2 mg/L vs. 0.1 mg/L). However, plasma copper concentrations were below the reference interval in most all calves in the study with only 2 of 17 having normal values (Table 3). The plasma activity of GGT and AST enzymes remained within the reference interval in all calves. Only plasma activity of the GD enzyme was, on average, higher than the reference interval for all calves (Table 3). Hepatic copper concentrations were, on average, within reference interval (Table 3).

## 4. Discussion

In the present study, we observed that treatment of milk alone was not enough to prevent calves from becoming MAP-shedders. The rate of positive fecal culture results was not different regardless of whether calves were fed untreated milk, copper-treated milk, calf milk replacer, or UHT-treated milk for human consumption. All experimental groups showed MAP shedding in feces at least once during the follow-up period, although it was intermittent. This indicates that despite the measures taken to prevent infection transmission through milk to calves, all methods failed.

Exposure of calves to MAP could have occurred through hay or silage, since MAP was isolated from these feeds. It is known that MAP exposure of cattle through fecal-contaminated forages can occur through slurry application to the crop fields or through fomites [29]. The survival of MAP in cattle slurry (pH 8.5 and 7% dry matter) has been estimated in 252 days at 5 °C and 98 days at 15 °C [30]. In addition, poorly fermented silages with high pH and low lactic acid content have been proposed as a risk factor for Johne’s disease [31].

Although we monitored the calves for one year, a longer study observation period could have revealed more significant differences among groups. In addition, monthly fecal sampling may have missed some intermittent MAP shedding [32]. Nevertheless, in group A one calf fed untreated naturally MAP-contaminated milk shed MAP in feces earlier than calves in other groups. This agrees with Mitchell et al. [33], who concluded that natural infections may have a much shorter time to shedding than generally assumed. On the other hand, the presence of MAP in fecal samples does not necessarily imply an active infection but could also be a transient passage of the bacteria through the intestine [34].

All calves, with the exception of one from group A, were ELISA-negative through the entire study period. This was not surprising, as the main limitation of the antibody ELISA is its inability to detect early stages of infection [35]. The presence of serum anti-MAP antibodies in one group A calf from the fifth month of study onward indicates an active immune response to a MAP infection, but since the assay does not have 100% specificity it could also have been a false-positive.

Contrary to expectations, the culture of milk samples for MAP was unrewarding since we only had one positive result. The use of chemical decontamination prior to culture might have had a detrimental effect on MAP recovery, as reported by others [36]. This may have been particularly critical in the present study where the MAP concentration in milk was not great, i.e., at or below the limit of the detection.

Copper ion treatment of milk significantly reduced the number of MAP. For 10 of 12 copper-treated milk samples, MAP counts were reduced to undetectable levels. However, this was not enough to prevent MAP infection transmission to calves in group B.

The detection limit of the BACTEC-MGIT system (10 bacterial MAP cells) on necropsy tissues may have been too high to detect the low numbers of MAP as others have reported [37,38]. This is supported by the observation of visible lesions consistent with MAP infection in calves fed untreated and copper-treated milk. These paucibacillary infections may have resulted from ingestion of low numbers of MAP in agreement with Mitchell et al. [33], who concluded that the initial MAP dose is the most important factor affecting the rate of infection progression. Mortier el al. [39] also observed that in young calves, a high inoculation dose resulted in more pronounced lesions than a low inoculation dose. Alternatively, these paucibacillary MAP lesions were merely a function of time, i.e., had the animals been necropsied at 3–5 years of age, the number of MAP in tissues may have been greater.

No evidence of copper toxicity was found during this study based on the plasma activity of the liver enzymes evaluated. Hepatic copper concentrations were also normal.

## 5. Conclusions

It was not enough to try to block MAP infection transmission, by only controlling the quality of milk consumed by calves. To be successful, paratuberculosis control programs must use multiple methods to interrupt infection transmission between the source, MAP-infected adult cows, and the most susceptible animals on the farm, young calves.

## Figures and Tables

**Figure 1 animals-11-02569-f001:**
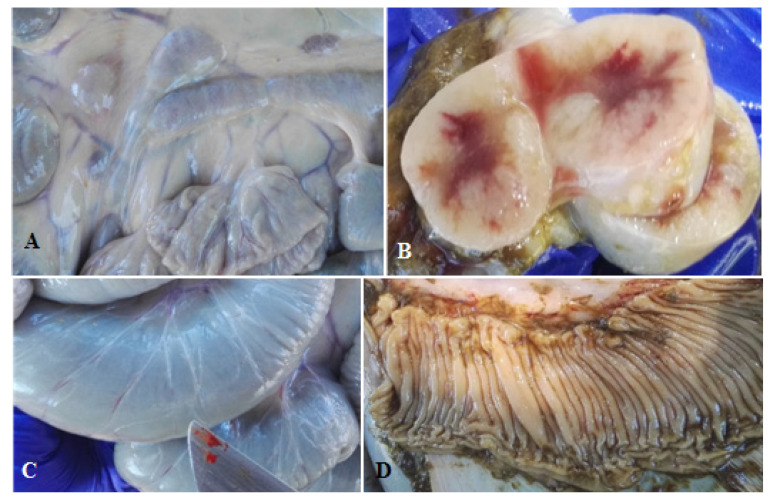
Findings observed after post-mortem examination of the calves. (**A**) Moderately enlarged mesenteric lymph nodes. Calf #050917, group A; (**B**) mesenteric lymph node with marked proliferation of cortical lymphoid tissue. Calf #300917, group A; (**C**) dilatation and signs of lymphangitis on the serosa of the ileum. Calf #300917, group A; and (**D**) folding of the ileal mucosa. Calf #011017, group A.

**Table 1 animals-11-02569-t001:** Estimated MAP concentration in naturally contaminated milk samples determined by qPCR before and after treatment with copper ions. Data represent number of MAP bacterial cell equivalents per mL of milk (centrifuged at 2500× *g* and resuspended in 1 mL of PBS). Calf milk replacer and UHT milk were also tested but no MAP bacteria were detected.

	Naturally Contaminated Milk (MAP bce/mL)	Copper-Treated Milk (MAP bce/mL)
	7.8 × 10^4^	5.7 × 10^4^
	3.3 × 10^4^	ND *
	5.66 × 10^4^	1.86 × 10^4^
	1.65 × 10^4^	ND *
	6.46 × 10^4^	ND *
	5.63 × 10^4^	ND *
	3.80 × 10^4^	ND *
	4.83 × 10^4^	ND *
	2.59 × 10^4^	ND *
	4.31 × 10^4^	ND *
	6.70 × 10^2^	ND *
	1.42 × 10^3^	ND *
Mean	3.85 × 10^4^	6.3 × 10^3^
	7.8 × 10^4^	5.7 × 10^4^
	3.3 × 10^4^	ND *
	5.66 × 10^4^	1.86 × 10^4^
	1.65 × 10^4^	ND *
	6.46 × 10^4^	ND *
	5.63 × 10^4^	ND *
	3.80 × 10^4^	ND *
	4.83 × 10^4^	ND *
	2.59 × 10^4^	ND *
	4.31 × 10^4^	ND *
	6.70 × 10^2^	ND *
	1.42 × 10^3^	ND *
Mean	3.85 × 10^4^	6.3 × 10^3^

* ND: not determined.

**Table 2 animals-11-02569-t002:** Fecal shedding pattern of MAP in the 4 groups of calves during the 12-month study. A solid dark gray box indicates a positive fecal culture confirmed by IS900-specific qPCR; a white box indicates a negative culture. Time after challenge (months).

Experimental Group	Calf ID	1	2	3	4 *	5	6	7	8	9	10	11	12
Contaminated milk (A)	270817												
Contaminated milk (A)	50917												
Contaminated milk (A)	300917												
Contaminated milk (A)	11017												
Contaminated milk (A)	21017												
Copper-treated milk (B)	311017												
Copper-treated milk (B)	11117												
Copper-treated milk (B)	21117												
Copper-treated milk (B)	31117												
Copper-treated milk (B)	41117												
Milk replacer (C)	31017												
Milk replacer (C)	41017												
Milk replacer (C)	51017												
Milk replacer (C)	61017												
UHT-milk (D)	121017												
UHT-milk (D)	131017												
UHT-milk (D)	141017												

* Beginning of silage feeding. No cultures reported as contaminated.

**Table 3 animals-11-02569-t003:** Mean plasma copper (Cu) concentrations, plasma activity of liver enzymes glutamate dehydrogenase (GD), gamma-glutamyl transferase (GGT), and aspartate aminotransferase (AST) and hepatic copper concentrations, per experimental group during the experimental period (mean ± standard deviation of 9 monthly samples). The reference intervals are shown in parentheses.

		Groups		
Blood Analyte	Contaminated Milk	Copper-Treated Milk	Milk Replacer	UHT Milk
GGT (3–39 U/L)	21 ± 2.6	21.3 ± 1.8	24.9 ± 3.7	27.0 ± 4.4
GD (2–28 U/L)	34 ± 4.3	35.4 ± 23	38.0 ± 14.3	58 ± 19.6
AST (25–125 U/L)	94.5 ± 12.1	91.6 ± 6.3	101.9 ± 14.1	104.0 ± 2.5
Plasma Cu (10–22 µmol/L)	9 ± 1.5	7.5 ± 1.8	8 ± 1.4	8.7 ± 1.5
Hepatic Cu (25–100 ppm)	25.4 ± 8.2	43.8 ± 17	45 ± 9.5	46.3 ± 20.4

## Data Availability

The datasets used and/or analysed during the current study are available from the corresponding author on reasonable request.

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
