# Peer review of "Attempted Control of Paratuberculosis in Dairy Calves by Only Changing the Quality of Milk Fed to Calves"

_animals, 2021, doi:10.3390/ani11092569_

Round 1

Reviewer 1 Report

The authors present an interesting paper, well written and well presented. 

Author Response

Reviewer I:

The authors present an interesting paper, well written and well presented. 

Response: we thank the reviewer and appreciate her/his comment

Reviewer 2 Report

The manuscript described an animal experiment of feeding calves on treated or untreated MAP positive milk samples. The important take home message that MAP free milk is not enough to stop the spread of the microorganisms  and other hygienic measure is a necessary. The manuscript is well written and deserved publication in the current form. I have only one remarks that the PCR was performed after culture. I think it is better to do it directly on the sample itself before cultivation. 

Author Response

REVIEWER II

The manuscript described an animal experiment of feeding calves on treated or untreated MAP positive milk samples. The important take home message that MAP free milk is not enough to stop the spread of the microorganisms  and other hygienic measure is a necessary. The manuscript is well written and deserved publication in the current form. I have only one remarks that the PCR was performed after culture. I think it is better to do it directly on the sample itself before cultivation. 

Response: we thank the reviewer and appreciate her/his comment. The PCR was run both as a confirmation of positive culture as well as directly from milk.

Reviewer 3 Report

The discussion on the use of copper-based treatment should be complemented (limitations, etc.) It should be noted if no histopathological lesions were observed in animals that received this milk.

Author Response

REVIEWER III

The discussion on the use of copper-based treatment should be complemented (limitations, etc.) It should be noted if no histopathological lesions were observed in animals that received this milk

Response: after the edits made in the present revised version, we think that the discussion covers what the reviewer point out as concern.

Reviewer 4 Report

General comments

Although control of paratuberculosis is an important research aspect, the study design employed here is poor, Materials and methods are not fully describe and the discussion is rather shallow and interpretation of results should be reconsiderd.

Specific comments

Lines 74-75: Papers about the efficacy of pasteurization are not recent.

Lines 83-84: The aim of the study described here is not the same one, which is described in abstract, and Introduction.

Line 88: 4L milk/once day should be changed to 4L milk/day

Lines 94-95: Was this based on statistical analysis? How could the authors define the minimum number, which gives trustworthy results?

Line 109: This is not a complete description of the experimental farm. Did the farm have already animals before begin of the study? Were these animals tested for MAP before the study animals arrive?

Line 114: Should also Silage be included here? Where did the described feedstuff come from?

Line 120: “minimum than” is improper English. Please consider revising.

Line 127: “transferred to the experimental farm” This is very important. Why did the authors decide to bring the nurse cows to the experimental farm? The cows are source of infection for calves of all groups. Have the cows had contact with some or all calves' groups?

Line 144: Was this for all milk groups?

It is also not clear if the milk in each group was collected from cows/purchased only once? If so how was it stored?

If the commercial milk replacer or milk was also sampled 12 times, could the authors provide a reason for that?

Lines 157-160: This should be considered as bias since the HPC would probably further reduce the number of viable MAP in milk. All milk groups should be treated equally (in the culturing procedure) in order to be able to compare between the efficacy of the different treatment approaches.

Line 163: Which IS900 PCR was used (published one?)

Line 188: does it make sense to use ELISA for MAP diagnosis in newborn calves, since the immune response of the MAP is usually a late response?

Line 210: were that the same blood sample used for ELISA?

Lines 250-251: This was only in one calf. Could it be simply due to pass over in the GIT without infection?

Lines 271-272: Is this not a redundancy of what was mentioned in lines 236-238?

Lines 305-307: one calf?

Lines 325-327: Is less than 10 MAP cells able to establish infection in tissues?

Author Response

REVIEWER IV

Specific comments

Lines 74-75: Papers about the efficacy of pasteurization are not recent 

Response: the mentioned references correspond to the first studies in relation to this topic. These references are relevant citations since they have become classic studies to account for this problem. We prefer to keep them.

Lines 83-84: The aim of the study described here is not the same one, which is described in abstract, and Introduction. 

Response: the reviewer is correct. The wording of the study objective has been standardized. Page (P) 3, Lines (L) 88-90, new version (NV).

Line 88: 4L milk/once day should be changed to 4L milk/day 

Response: the reviewer is correct. It has been changed as suggested. P3 L93 NV.

Lines 94-95: Was this based on statistical analysis? How could the authors define the minimum number, which gives trustworthy results? 

Response: This was not based on a statistical criterion, but rather on a criterion that considered animal welfare concern, since calves were exposed-infected and euthanized by the end of the experiment, suitable facilities as well as economical aspect.

Line 109: This is not a complete description of the experimental farm. Did the farm have already animals before begin of the study? Were these animals tested for MAP before the study animals arrive? 

Response: the farm kept some cows for milk production, which were grazing far and totally separated from where the calves were grouped and never had contact with them. In addition, as mentioned in the paper, technical staff entered the pens rigorously wearing clean coveralls, boots, and gloves exclusively for each pen, as well as they also used exclusive items to keep the pen clean to eliminate the risk of cross contamination.

Line 114: Should also Silage be included here? Where did the described feedstuff come from? 

Response: as stated in P3 L122 NV, grass silage was offered from 12 weeks onwards. Information about feedstuff origin have been included for clarity P3 L122-124 NV.

Line 120: “minimum than” is improper English. Please consider revising. 

Response: the reviewer is correct. The sentence has been re-written for clarity. P3 L129 NV.

Line 127: “transferred to the experimental farm” This is very important. Why did the authors decide to bring the nurse cows to the experimental farm? The cows are source of infection for calves of all groups. Have the cows had contact with some or all calves' groups? 

Response: the nurse cows were brought to the farm for practical reasons, basically to be milked every day to feed calve groups A and B. These nurse cows were not in contact with the calves at any time, so the risk as a source of infection of transmitting MAP by a route other than milk, was therefore non-existent.

Line 144: Was this for all milk groups? 

Response: yes, we referred to all groups of calves that received milk or milk replacer. The sentence has been re-written for clarity. P4 L156 NV.

It is also not clear if the milk in each group was collected from cows/purchased only once? If so how was it stored? 

Response: naturally contaminated milk was obtained from nurse cows and fed to calves immediately or after copper treatment, and it was not stored. The UHT milk was stored in a proper storeroom and the milk replacer was reconstituted when feeding the calves.

If the commercial milk replacer or milk was also sampled 12 times, could the authors provide a reason for that? 

Response: all the milks were sampled during the 12 weeks dairy diet, so they were all sampled the same number of times, in order to be able to compare the results.

Lines 157-160: This should be considered as bias since the HPC would probably further reduce the number of viable MAP in milk. All milk groups should be treated equally (in the culturing procedure) in order to be able to compare between the efficacy of the different treatment approaches 

Response: the basic protocol for the isolation of MAP from milk includes centrifugation to concentrate bacterial cells, chemical decontamination (often using hexadecylpyridinium chloride [HPC]) to decrease contaminating microorganisms, and culture in either liquid or solid medium optimized for the recovery of MAP. Although HPC is used to control contaminating microorganisms, even though this decontaminant may exert some negative effect on MAP, it is accepted that this should be minimal.

Line 163: Which IS900 PCR was used (published one?) 

Response: the IS900 PCR protocol used was the one cited as reference [22]. P4 L175 NV.

Line 188: does it make sense to use ELISA for MAP diagnosis in newborn calves, since the immune response of the MAP is usually a late response? 

Response: the reviewer is correct. However, since it has been reported that in experimental MAP infection in calves, humoral immune response has been informed, then we wanted to see what happened to the immune response of the calves under the conditions of the present study.

Line 210: were that the same blood sample used for ELISA? 

Response: yes, was the same blood sample as the one used for the ELISA test

Lines 250-251: This was only in one calf. Could it be simply due to pass over in the GIT without infection?  Response: the reviewer is correct, it was just one calf, and the transient passage of MAP through the intestine is mentioned in the discussion. P9 L379-380 NV.

Lines 271-272: Is this not a redundancy of what was mentioned in lines 236-238? 

Lines 305-307: one calf? 

Response: the reviewer is correct. The sentences have been re-written for clarity. P6 L253-256 and P9 L375-377 NV.

Lines 325-327: Is less than 10 MAP cells able to establish infection in tissues? 

Response: so far, there is no evidence of the number of minimal MAP cells that can successfully establish an active infection. What arises in the discussion is that the analytical sensitivity of the culture system used could be one of the explanations for why MAP could not be cultured from tissue.